Using molecular characteristics to distinguish multiple primary lung cancers and intrapulmonary metastases

Li Zhenhua 1
Lv Huilai 1
Zhang Fan 1
Zhu Ziming 2
Guo Qiang 3
Wang Mingbo 1
Huang Chao 1
Guo Lijie 4
Meng Fanfei 4
Tian Ziqiang 1 tianzq1026@163.com
1 Department of Thoracic Surgery, Fourth Hospital of Hebei Medical University , Shijiazhuang , China
2 Department of Thoracic Surgery, The First Hospital of Xingtai , Xingtai , China
3 Department of Thoracic Surgery, Affiliated Hospital of Hebei University , Baoding , China
4 OrigiMed , Shanghai , China
Guan Fanglin
Electronic publication date: 2024 Jan 31
Publication date: 2024
Volume: 12
Electronic Location ID: e16808
Received 2023 Oct 3; Accepted 2023 Dec 29
Copyright: © 2024 Li et al.
Copyright year: 2024
Copyright holder: Li et al.
License: This is an open access article distributed under the terms of the Creative Commons Attribution License, which permits unrestricted use, distribution, reproduction and adaptation in any medium and for any purpose provided that it is properly attributed. For attribution, the original author(s), title, publication source (PeerJ) and either DOI or URL of the article must be cited.
License URL: https://creativecommons.org/licenses/by/4.0/

Keywords: Multiple lung cancer, Multiple primary lung cancer, Intrapulmonary metastasis, Next-generation sequencing, Molecular classification

Funding: The authors received no funding for this work.

==============================
Objectives

Multiple lung cancers may present as multiple primary lung cancers (MPLC) or intrapulmonary metastasis (IPM) with variations in clinical stage, treatment, and prognosis. However, the existing differentiation criteria based on histology do not fully meet the clinical needs. Next-generation sequencing (NGS) may play an important role in assisting the identification of different pathologies. Here, we extended the relevant data by combining histology and NGS to develop detailed identification criteria for MPLC and IPM.

Materials and Methods

Patients with lung cancer (each patient had ≥2 tumors) were enrolled in the training (n = 22) and validation (n = 13) cohorts. Genomic profiles obtained from 450-gene-targeted NGS were analyzed, and the new criteria were developed based on our findings and pre-existing Martini & Melamed criteria and molecular benchmarks.

Results

The analysis of the training cohort indicated that patients identified with MPLC had no (or <2) trunk or shared mutations. However, 98.02% of mutations were branch mutations, and 69.23% of MPLC had no common mutations. In contrast, a higher percentage of trunk (33.08%) or shared (9.02%) mutations were identified in IPM, suggesting significant differences among mutated components. Subsequently, eight MPLC and five IPM cases were identified in the validation cohort, aligning with the independent imaging and pathologic distinction. Overall, the percentage of trunk and shared mutations was higher in patients with IPM than in patients with MPLC. Based on these results and the establishment of new determination criteria for MPLC and IPM, we emphasize that the type and number of shared variants based on histologic consistency assist in identification.

Conclusion

Determining genetic alterations may be an effective method for differentiating MPLC and IPM, and NGS can be used as a valuable assisting tool.

Introduction

Lung cancer has the highest mortality rate of all malignancies worldwide (Martini & Melamed, 1975; Siegel, Miller & Jemal, 2020). Patients with a history of lung cancer or those with biopsy-proven synchronous lesions are suspected of having multiple lung cancers. Even though multiple primary lung cancers are common (8.4–10% of the diagnosed cases), most of the synchronous lesions indicate metastases (Kozower et al., 2013; Mansuet-Lupo et al., 2019; Nakata et al., 2004). Therefore, it is critical to determine whether the multiple lung cancers are intrapulmonary metastases (IPM) or multiple primary lung cancers (MPLC, synchronous or metachronous occurrences). They represent different TNM stages, therapeutic modalities, prognostic endpoints, and clonal origins. The eighth edition of the TNM staging system for lung cancer classifies multiple nodules within the same lobe as T3, different but ipsilateral lobes as T4, and contralateral lobes as M1 based on the assumption that multiple nodules are IPM (Goldstraw et al., 2016). This staging rationale prevents surgical resection for curing the disease and indicates palliative care even for patients with MPLC. The occurrence of two or more primary malignant tumors in the lungs of the same patient without N2/N3 lymph nodes or systemic metastasis is termed MPLC (Liu et al., 2019). Patients with IPT and IPM have different stages and require different treatment strategies (Chen et al., 2018; Fabian et al., 2011). Patients with MPLC can often achieve long-term survival with radical surgery, whereas patients with IPM require aggressive systemic or even palliative treatment (Jiang et al., 2015). IPM tumors represent the same clonal origin, whereas MPLCs have different clonal origins. Notably, the driver genes are different between these two entities (Rodriguez et al., 2020). However, most MPLCs are misdiagnosed as IPM based on pathologic and radiologic findings (Shen et al., 2015), and distinguishing MPLC from IPM in multifocal lung cancer remains a common clinical challenge.

Martini & Melamed (1975) proposed clinical differential criteria in 1975 to distinguish MPLC and IPM. These practically valuable criteria primarily focused on the histologic characteristics of tumors. However, the criteria are rather empirical and difficult to implement when histologic features are similar (Martini & Melamed, 1975). The TNM staging system of multifocal lung cancers in the seventh edition of the AJCC staging manual is mainly based on the Martini & Melamed criteria (Martini & Melamed, 1975; Shen et al., 2007). The comprehensive histologic assessment adopted by the World Health Organization in 2015 remains the standard for differentiating MPLC from IPM in case of multiple pulmonary nodules (Travis et al., 2015). The NCCN guidelines recommend a multidisciplinary approach to diagnosing multiple lung cancer, which includes thoracic radiology, pulmonary medicine, thoracic surgery, medical oncology, and radiation oncology. However, only histologic classification and clinical data are not sufficient to distinguish MPLC from IPM. A metareview indicated that histology has a moderate differential diagnostic value between MPLC and IPM (Tian et al., 2022).

Molecular profiling may help determine whether different lung nodules are clonally related. Some research teams have proposed molecular identification criteria for distinguishing MPLC from IPM (Mansuet-Lupo et al., 2019; Pei et al., 2021; Wang et al., 2020). Mansuet-Lupo et al. (2019) supplemented the histologic classification with the next-generation sequencing (NGS) analysis of 22 hotspot genes to determine the concordance between the histologic and genomic classifications using the K-test. Inconsistent cases were reclassified using a combinatorial histomolecular algorithm. However, the interpretation of samples without hotspot mutations was neglected because the algorithm was restricted to a limited number of hotspot genes in lung cancer. Wang et al. (2020) increased the number of tested genes from 22 to 605 and differentiated MPLC and IPM based on the number of overlapping genes in multiple samples from a patient. Patients with no overlapping alterations between multiple samples or with only one overlapping high- or low-frequency driver alteration or both were interpreted as MPLC. IPM was interpreted only if more than three common mutations were found in all the samples. However, the authors did not provide a definitive explanation for patient samples in which mutations could not be detected. Pei et al. (2021) used an 808-gene panel to perform additional discrimination beyond histology between MPLC and IPM samples based on the discriminatory power of the EGFR L858R mutation. MPLC was identified when the EGFR L858R mutation was the only consistent mutation in the lung tumors of the patient, and IPM was identified when the same driver mutation (excluding EGFR L858R) was present in all tumors or when all the alterations were common to the tumors of the patient. The authors emphasized that the tumor cannot be classified if mutations are not detected in a tumor lesion. Overall, these studies provided a comprehensive molecular approach to differentiate MPLC and IPM when histology was not sufficient for differentiation. However, generally accepted molecular identification criteria are not available for clinical use.

In our study, we created the training and validation cohorts comprising patients with multiple adenocarcinoma lesions and conducted a 450-gene NGS analysis. We proposed the updated molecular criteria based on the pre-existing molecular identification criteria for MPLC and IPM and our findings. We grouped samples based on histologic concordance between samples and then analyzed the grouping in the case of different overlapping genetic variants or even in the case of undetected genetic variants in one or more samples. This criterion focuses on identifying common mutations and extends the discussion on rare variants. Taken together, analyzing genetic alterations may be an effective method to differentiate MPLC and IPM, and NGS may be a useful supportive tool.

Materials and Methods

Patient selection

We included 35 patients having more than one lung carcinoma in the study. All patients visited the Fourth Hospital of Hebei Medical University between December 2014 and April 2020 and sufficient specimens were obtained for histologic and molecular analysis. Patients who received neoadjuvant therapies and those with extrathoracic metastasis were excluded from the study. All experimental plans and protocols were submitted to the ethics/licensing committees of the participating hospitals for thorough review and approval before conducting the retrospective study. The protocols were approved by the Fourth Hospital of Hebei Medical University (2018MEC133).

Sample preparation, targeted NGS, and genetic analysis

All tumor tissues and matched blood samples were tested for genomic targeting based on NGS of 450 genes (Cancer Sequencing YS panel, CSYS; OrigiMed, Shanghai, China; (Cao et al., 2019)). Formalin-fixed paraffin-embedded (FFPE) tumor tissues and matched blood samples were collected from the Fourth Hospital of Hebei Medical University. As previously described in Cao et al. (2019), genetic testing is performed in Origimed laboratory, which is accredited both by CAP (College of American Pathologists) and CLIA (Clinical Laboratory Improvement Amendments). The histologic subtyping was reviewed by independent pathologists with 10 years of experience from Origimed. The percentage of tumor cells in each sample was ≥20%. Approximately 50 ng DNA was extracted from 40-mm FFPE tumor tissues and blood samples using the QIAamp DNA FFPE Tissue Kit (Qiagen, Hilden, Germany) for subsequent genetic analysis. All the coding exons of 450 cancer-related genes and the selected introns of the parts of the targeted genes that frequently rearranged in solid tumors were captured using the custom hybridization capture panel, and DNA sequencing libraries were constructed. The libraries were sequenced on an Illumina NextSeq-500 Platform (Illumina Incorporated, San Diego, CA, US). The sequencing depth mean coverage was 1,000× and 300× for FFPE and matched blood samples, respectively. The CSYS panel was used to determine exon sequences encoding all of the 450 genes and the TERT promoter and introns of 39 genes (Cao et al., 2019). The identified somatic alterations contained mutations and copy number changes. MuTect, Pindel, and EXCAVATOR were used to identify SNVs, InDels, and CNVs, respectively. A definite SNV/InDel required a minimum of five unique supporting reads. An in-house developed algorithm was used to screen gene rearrangements. All variants were manually viewed on Integrative Genomics Viewer to avoid misreporting (Cao et al., 2019). We emphasized that the genes included in the atlas were variants with identified or potentially important functional changes as annotated by the OncoKB and ClinVar databases (Chakravarty et al., 2017; Landrum et al., 2020). In addition, we excluded gene variants with unknown functional changes, no predicted important functional changes, or unreported undetermined changes to avoid inter-observer inconsistencies in the interpretation of gene mutations.

Study design

Our retrospective cohort study included patients with MPLC and IPM adenocarcinoma. The primary inclusion criteria were adults aged ≥18 with comprehensive clinicopathologic information and confirmed diagnosis of lung adenocarcinoma based on imaging and pathologic analysis. Sufficient tumor tissue was obtained from multiple lesions for NGS sequencing. Patients with incomplete information, lung metastases other than lung cancer, and the absence of tissue or control samples for patients with primary exclusion criteria included suspicion.

Finally, 35 patients with complete sample details and clinical information (21 MPLC and 14 IPM) were included in the study. The training and validation cohorts were established in the chronologic order of enrolment in a 2:1 ratio to avoid bias. Twenty-two patients were enrolled in the training cohort, and the remaining 13 patients were included in the validation cohort. The determination of MPLC or IPM by clinicians was based on the IASLC guidelines for the classification of lung cancers with multiple pulmonary sites of involvement (before conducting genetic tests). We analyzed the imaging findings, anatomical evidence, pathologic findings, histologic examination of the primary tumor, and histologic subtypes to differentiate MPLC from IPM. Two pathologists independently examined all biopsies, and consensus was achieved for all samples. We used the Martini & Melamed criteria to classify these samples into MPLC and IPM. Synchronous tumors located in different segments but showing similar histologic types without lymphatic or systemic metastases were classified as MPLC. In addition, samples with different histologic types, at different segmental or lobar locations, with different growth rates and biomarker patterns, and without lymph node or lung cancer metastases were classified as MPLC. Heterochronous tumors were classified as MPLC when the time interval between tumors was >2 years. Such tumors were classified as IPM when the time interval between tumors was <2 years and both tumors were located in the same lobe or when the tumor interval was <2 years and the tumors were located in different lobes of the lungs with lymph node involvement or systemic metastases. Doctors assessed the available evidence and interpreted whether the tumors were primary lesions or primary tumors with metastasis.

Statistical analysis

Student’s t-test was used to compare two groups, whereas ANOVA and post hoc tests (mainly the Bonferroni test) were used to compare three or more groups. The Chi-square and Fisher tests were used to determine the significance of rates or percentages when required. A p-value of less than 0.05 was considered statistically significant.

Results

Clinical characteristics of patients

Patients with >2 lung lesions were divided into training (n = 22) and validation (n = 13) cohorts. The median age of the training cohort was 56.5 years (range: 43–77 years) and that of the validation cohort was 57 years (range: 27–75 years). The proportion of females was 40.9% and 53.8% in the training and validation cohorts, respectively, and the proportion of patients with adenocarcinomas was 91.7% and 100% in the training and validation cohorts, respectively. Twenty-nine patients had two tumors, five patients had three tumors, and one patient had four tumors; therefore, 77 individual tumors were analyzed. Adenocarcinoma (n = 45), invasive adenocarcinoma (IAC; n = 3), adenocarcinoma in situ (AIS; n = 10) and minimally invasive adenocarcinoma (MIA; n = 6) were the main pathologic types. Among the 77 tumors, 17, 16, 15, and 25 tumors were stage I, II, III, and IV tumors, respectively. Twenty-two patients in the training cohort included nine patients with MPLC and 13 with IPM, and 13 patients in the validation cohort included eight patients with MPLC and five with IPM (Table 1).

Table 1 Clinicopathological information for lung cancer patients involved in this study.

Patient characteristics	Training cohort
(n = 22)	Validation cohort
(n = 13)	
Sex, n (%)			
Male	13 (59.1%)	6 (46.2%)	
Female	9 (40.9%)	7 (53.8%)	
Age (year), y (range)			
Median	57	56.5	
Range	27–75	43–77	
≤60	13 (59.1%)	6 (53.8%)	
>60	9 (40.9%)	7 (46.2%)	
Tumor characteristics (n = 77)			
Stage, n (%)			
I	8 (16.7%)	9 (31.0%)	
II	7 (14.6%)	9 (31.0%)	
III	11 (22.9%)	4 (13.8%)	
IV	18 (37.5%)	7 (24.2%)	
unknown	4 (8.3%)	0 (0%)	
Histology, n (%)			
AIS	3 (6.2%)	7 (24.2%)	
MIA	1 (2.1%)	5 (17.2%)	
ADC	31 (64.6%)	14 (48.3%)	
IAC	0 (0.0%)	3 (10.3%)	
Others	13 (27.1%)	0 (0.0%)	
Lesion type			
MPLC	13 (59.09%)	8 (61.5%)	
IPM	9 (40.91%)	5 (38.5%)	
Note:

AIS: adenocarcinoma in situ; MIA: minimally invasive adenocarcinoma; IAC: invasive adenocarcinoma; ADC: adenocarcinoma.

Establishing the criteria for distinguishing MPLC from IPM

NGS was performed to detect mutational alterations in all lesions to explore strategies for differentiating MPLC and IPM. Overall, 510 mutations were detected in 48 tumors of 22 patients from the training cohort. The genes with high mutational frequency were EGFR (54%), TP53 (50%), MUC16 (22%), KRAS (17%), LRP1B (17%), ERBB2 (15%), RBM10 (15%), OBSCN (13%) and PIK3CA (13%), and the majority of mutations observed were substitution/deletion mutations (Fig. 1A). The substitution/indel and gene amplification of EGFR mutations (e.g., EGFR p. (E709A; G719S), EGFR p. T790M, EGFR p. E746_A750del, p. L858R, and EGFR amplification) were observed in six patients. Ten tumors from these six patients had the EGFR p. L858R mutation. Gene fusion was only found in ROS1, NTRK3, and RET among 22 patients. ROS1 fusion was observed in both the tumors of one patient. NTRK3 fusion was observed in one tumor, and RET fusion was found in one tumor from two patients (Table S1; ROSi fusion was observed in both the tumors of one patientNTRK3 fusion was observed in one tumor, and RET fusion wasfound in one tumor from two patients). Eight patients (36.4%) showed inconsistent driver mutations and unique mutation profiles for each tumor, and no mutation was detected in a tumor lesion of one patient (4.5%). The remaining 13 patients (59.1%) had one or more mutational alterations in each tumor (Table S1).

Figure 1 Mutation status and concordance rate for lesions of the training cohort.

(A) The mutational landscape of all lesions from the training cohort, including 13 MPLC and nine IPM patients. The top 20 mutated genes are shown as indicated. (B) The number of mutations for all lesions of all patients with MPLC. (C) The number of mutations for all lesions of all patients with IPM.

The Martini & Melamed criteria and previously proposed molecular identification methods (Mansuet-Lupo et al., 2019; Pei et al., 2021; Wang et al., 2020) were used to diagnose patients in the training cohort (Table 2). We found that the diagnoses of 14/22 (63.6%) patients were consistent with all previously published criteria. These 14 patients included 10 patients with MPLC, which had one or no common mutations, and four patients with IPM, which had more than three mutations (Tables S1 and S2). Patient 2, having three tumors, was identified as MPLC using the Martini & Melamed criteria but was identified as MPLC/IPM using different molecular methods (Table 2). In this patient, tumor 1 or 3 and tumor 2 had one common mutation (NTRK1 c.288-3C > A); tumors 1 and 3 had three common mutations (NTRK1 p. G137V, NTRK1 c.288-3C > A, and PIK3CA p. K111N). Therefore, tumor 2 was the primary lesion, and tumors 1 and 3 were metastatic lesions. Patients 4 and 7, having more than three common mutations, were identified as MPLC using the Martini & Melamed criteria but were identified as IPM using molecular methods. Patients 5, 6, 8, 12, and 15 were identified using the previously proposed molecular methods but had different classifications. Patient 5 (common mutations: ERBB2 c.1022-6C > T and ROS1 fusion) and patient 8 (common mutations: EGFR p. L858R and TP53 p. W146*) were identified as IPM based on the molecular methods proposed by Mansuet-Lupo et al. (2019) (IPM can be interpreted when tumors have at least two mutations in common) and Pei et al. (2021) (IPM can be identified when EGFR p. L858R and another mutation are shared between tumors or when consistent gene hotspot mutations (EGFR, KRAS, BRAF, ERBB2, ALK, ROS1, MET, or RET) occur in tumors, but EGFR p. L858R is not shared). However, MPLC was identified in these two patients based on the molecular methods proposed by Wang et al. (2020). This classification was supported by the identification of TKI-related driver gene alterations (EGFR, ALK, ROS1, MET, RET, BRAF, and KRAS) and non-TKI-related gene alterations with the total number of common mutations ≤2. Additionally, MPLC was confirmed by pathologic analysis and molecular identification. Two tumors of patient 12 had the common mutations PIK3CG p. H199N and POLD1 p. A532T. The patient was identified as IPM based on the methods proposed by Mansuet-Lupo et al. (2019) and Pei et al. (2021). However, the same patient was interpreted as MPLC based on the classification method proposed by Wang et al. (2020). We finally identified two lesions in this patient as MPLC based on the histologic types (AIS and MIA) and the common mutations.

Table 2 Patients with multiple lung cancer in training cohort classified by the Martini and Melamed criteria and molecular methods.

Patient No.	Martini &
Melamed
criteria	Mansuet-Lupo et al. (2019)	Wang et al. (2020)	Pei et al. (2021)	Final
diagnosis	
TP1	MPLC	MPLC	MPLC	MPLC	MPLC	
TP2	MPLC	IPM	IPM	IPM	IPM	
TP3	IPM	IPM	IPM	IPM	IPM	
TP4	MPLC	IPM	IPM	IPM	IPM	
TP5	MPLC	IPM	MPLC	IPM	IPM	
TP6	MPLC	MPLC	MPLC	IPM	MPLC	
TP7	MPLC	IPM	IPM	IPM	IPM	
TP8	MPLC	IPM	MPLC	IPM	MPLC	
TP9	MPLC	MPLC	MPLC	MPLC	MPLC	
TP10	MPLC	MPLC	MPLC	MPLC	MPLC	
TP11	MPLC	MPLC	MPLC	MPLC	MPLC	
TP12	MPLC	IPM	MPLC	IPM	MPLC	
TP13	MPLC	MPLC	MPLC	MPLC	MPLC	
TP14	IPM	IPM	IPM	IPM	IPM	
TP15	IPM	IPM	MPLC	IPM	IPM	
TP16	MPLC	MPLC	MPLC	MPLC	MPLC	
TP17	IPM	IPM	IPM	IPM	IPM	
TP18	MPLC	MPLC	MPLC	MPLC	MPLC	
TP19	IPM	IPM	IPM	IPM	IPM	
TP20	MPLC	MPLC	MPLC	MPLC	MPLC	
TP21	MPLC	MPLC	MPLC	MPLC	MPLC	
TP22	MPLC	MPLC	MPLC	MPLC	MPLC	

Mutations are divided into three categories, namely trunk, shared, and branch mutations (Wang et al., 2020). We examined the occurrence of all three types of mutations in 13 MPLC and 9 IPM specimens. The results revealed that 9/13 MPLCs exclusively had branch mutations, indicating that 69.23% of MPLC had no common mutations. This rate was similar to the previously reported rate of 66.7% (Wang et al., 2020). The remaining 4 MPLCs had <2 trunk mutations and no shared mutation, indicating a very low level of common mutations (Fig. 1B). Conversely, all IPM showed a significantly higher number of trunk or shared mutations, highlighting marked disparities among mutant components (Fig. 1C). Additionally, we demonstrated that branch mutations accounted for 98.02% of the mutations in MPLC compared with only 57.89% mutations in IPM. Trunk mutations constituted 33.08% and shared mutations accounted for 9.02% in IPM (Fig. S1A).

We created a flowchart of our criteria to distinguish MPLC from IPM based on our observations (Fig. 2). When the concordance rate is 0%, A-1) and there is no common mutation between the samples, then the patient is judged to be MPLC; A-2) and there are samples with no mutation, it is necessary to differentiate between the number of samples with no mutation; A-2.1) if all samples have no detectable mutation, then the relationship between the samples cannot be determined; A-2.2) if there are only two samples and one of the samples has no mutation and the other has a mutation, then the patient is the patient is judged to be MPLC; A-2.3) if there are >2 samples and one of the samples has no mutation and the remaining sample has a mutation, then the patient cannot be judged. When the concordance rate is >0%, B-1) and there is only one common mutation between the samples, then the patient is judged to be MPLC; B-2.1) if there are two common mutations between the samples (one of which is a driver mutation and the other is a rare mutation) then the patient is considered to be an IPM; B-2.2) if there are either two common driver mutations between the samples or one common driver mutation and one p53 mutation, then the patient is considered to be patient was MPLC; B-2.3) if it was the presence of two common rare mutations, the patient could not be determined; B-3) if there were more than three common mutations between samples or all mutations were common, the patient was considered to be IPM. Driver mutations are defined as mutations derived from EGFR, KRAS, BRAF, ERBB2, ALK, ROS1, NRAS, RET, or driver mutations in the MET gene.

Figure 2 Proposed algorithm for classifying MPLC and IPM based on molecular criteria.

Validation of interpretation criteria for the identification of MPLC and IPM

We created a validation cohort of 13 patients to validate our criteria and streamline the protocol for distinguishing MPLC and IPM. NGS was performed to detect mutational alterations in all lesions to differentiate between MPLC and IPM. Overall, 316 mutations were detected in 29 tumors. The genes with high mutational frequencies were EGFR (52%), TP53 (31%), KRAS (24%), LRP1B (24%), RBM10 (21%), CFTR (17%), and CNAS (17%). These findings were similar to those observed in the training group (Fig. 3A). Moreover, EGFR mutations (EGFR p. L858R, EGFR cn_amp, and EGFR p. E746_A750del) were observed in four patients, and eight tumors of these four patients had EGFR p. L858R mutation. RB1 fusion was observed in both tumors of one patient. The EGFR, CDK4, GNAS, NPM1, SDHA, TRIO, ZNF217, and NFKBIA gene amplifications were identified in tumor samples from paired tumors of three patients. Six patients (46.2%) showed inconsistent driver mutations and unique mutation profiles in each tumor. The other seven patients (53.8%) had one or more mutational alterations in each tumor (Table S2). Five patients were interpreted as IPM, and the remaining seven patients appeared to be MPLC (Table 3). Subsequent analysis of mutational status and genes involved supported this interpretation. Figure 3B shows that six out of eight MPLCs do not have any trunk or shared mutation but only branch mutations. Therefore, 75% of MPLC did not have any common mutations, and these findings were similar to those observed in the training cohort. The remaining two MPLCs had only one trunk mutation, indicating a very low occurrence of common mutations. In contrast, IPM had a significantly higher number of trunk mutations, indicating notable differences among mutant components (Fig. 3C). The number of trunk mutations in IPM was higher and that of branch mutations was lower compared with MPLC (Fig. S1B). Overall, these findings confirmed our criteria for distinguishing MPLC from IPM.

Figure 3 Mutation status and concordance rate for lesions of the validating cohort.

(A) The mutational landscape of all lesions from the validating cohort, including eight MPLC and five IPM patients. The top 20 mutated genes are shown as indicated. (B) The number of mutations for all lesions of all patients with MPLC. (C) The number of mutations for all lesions of all patients with IPM.

Table 3 Patients with multiple lung cancer in validating cohort classified by the Martini and Melamed criteria and molecular methods.

Patient No.	Martini &
Melamed
criteria	Mansuet-Lupo et al. (2019)	Wang et al. (2020)	Pei et al. (2021)	Final
diagnosis	
VP1	IPM	IPM	MPLC	IPM	IPM	
VP2	IPM	IPM	IPM	IPM	IPM	
VP3	IPM	IPM	IPM	IPM	IPM	
VP4	MPLC	MPLC	MPLC	MPLC	MPLC	
VP5	MPLC	MPLC	MPLC	MPLC	MPLC	
VP6	IPM	IPM	IPM	IPM	IPM	
VP7	MPLC	MPLC	MPLC	MPLC	MPLC	
VP8	MPLC	MPLC	MPLC	MPLC	MPLC	
VP9	MPLC	MPLC	MPLC	IPM	MPLC	
VP10	MPLC	MPLC	MPLC	MPLC	MPLC	
VP11	MPLC	MPLC	MPLC	MPLC	MPLC	
VP12	MPLC	MPLC	MPLC	MPLC	MPLC	
VP13	IPM	IPM	IPM	IPM	IPM	

Comprehensive analysis of the EGFR alterations

Alterations in the EGFR gene were identified in 21/35 patients (60%) enrolled in the study i.e., 38 lesions from 21 patients showed this mutation. Sixteen MPLC lesions showed EGFR mutations. Eight of them had exon 21 L858R, two had EGFR p. A767_V769dup, and the remaining six presented with EGFR p. (E709A; G719S), EGFR p. A750_I759delinsGD, EGFR p. G719S, EGFR p. L833V, EGFR p. E421Q, EGFR p. H358L, and EGFR amplification. Twenty-three IPM lesions showed EGFR alterations. Twelve lesions showed EGFR p. L858R, eight presented with EGFR p. E746_T751delinsA, two lesions showed EGFR p. L747_T751del, four lesions showed EGFR p. T790M, and nine lesions showed EGFR amplification (Table S1). In addition, we analyzed the number of EGFR, KRAS, TP53, BRAF, ERBB2, MET, RET, NRAS, ALK, and ROS1 mutations in patients identified with MPLC and IPM and found that KRAS mutations were significantly higher in MPLC (42.86%) than in IPM (7.14%; Table S3).

Molecular methods and martini criteria for identifying MPLC and IPM

We used the Martini & Melamed criteria in addition to molecular methods to distinguish whether the ambiguous samples can be categorized as MPLC or IPM. We analyzed whether the identification results were consistent between the two modalities. The first patient was a 70-year-old Chinese woman admitted to hospital. The computed tomography (CT) examination showed multiple nodules in the right lung, and metastasis was not excluded. Two nodules were observed through the CT examination (Fig. 4A). T1 nodule, measuring 1.6 × 0.8 × 1 cm3, was observed in the apical segment of the upper lobe of the right lung, and T2 nodule, measuring 1.2 × 0.6 × 0.5 cm3, was observed in the middle lobe of the right lung. Pathologic examination of these two pulmonary nodules indicated IAC in the lung tissue. Lepidic was the dominant type in the T1 nodule, and the T2 nodule was 30% acinar and 70% lepidic. The NGS results revealed that these two lesions have only one common mutation EGFR p. L858R, indicating MPLC. The second patient, a 69-year-old Chinese lady, was initially diagnosed with only one lesion was found in the right middle lobe on CT. Later, another lesion was found on surgery, indicating metastases. T1 nodule, measuring 5 × 3.5 × 3 cm3, was observed in the middle lobe of the right lung, and T2 nodule, approximately 0.8 cm in diameter, was observed in the upper lobe of the right lung. Pathologic examination of the two pulmonary nodules indicated IAC in the lung tissue, and both nodules were 50% acinar, 30% papillary, and 20% mucinous. The NGS results revealed two common mutations CDK4 amplification and EGFR p. L858R (concordance rate >50%), indicating IPM (Fig. 4B).

Figure 4 Radiologic and pathologic appearances and NGS profile of an MPLC of IAC with three nodules.

In addition, another two patients were identified through both pathologic examination and molecular methods. A 77-year-old Chinese woman was diagnosed with multiple nodules in the right lung, and metastasis was not excluded. Two nodules were observed through CT examination. T1 nodule in the upper lobe of the right lung was approximately 1 cm in diameter, and T2 nodule in the lower lobe of the right lung measured 1.5 × 1 × 1 cm3. Pathologic examination of the two pulmonary nodules suggested IAC in the lung tissue. The type of T1 nodule was 80% lepidic and 20% acinar, whereas the type of T2 nodule was 80% acinar and 20% lepidic. The NGS results revealed that these two lesions had no common mutation, indicating MPLC (Fig. 5A). The fourth patient was a 66-year-old Chinese man having three nodules in the upper lobe of the right lung. The CT examination showed two nodules (T1 and T2), both measuring 1 × 0.8 × 0.5 cm3, in the upper lobe of the right lung. T3 nodule in the upper lobe of the right lung measured 1 × 1 × 0.8 cm3. Pathologic examination of the three pulmonary nodules showed IAC in the lung tissue. The type of T1 nodule was 20% lepidic and 80% acinar, the type of T2 nodule was 90% acinar and 10% lepidic, and the type of T3 nodule was 30% acinar and 70% lepidic. The NGS results revealed that these two lesions had no common mutation, indicating MPLC (Fig. 5B). The molecular identification in these four patients was consistent with the pathologic diagnosis, suggesting that our molecular identification approach can be used to distinguish MPLC from IPM in the absence of pathologic diagnosis.

Figure 5 Radiologic and pathologic appearances and NGS profile of two patients of IAC with two nodules.

Discussion

Multiple lung nodules are increasingly being detected in patients with lung cancer with the widespread launch of large-scale cancer screening projects equipped with high-resolution imaging devices. The reported incidence of multiple lung nodules ranges from 5% to 20% in varying sample sizes and populations (Chen et al., 2019; Gazdar & Minna, 2009). These nodules need to be definitively identified as MPLC or IPM because the staging, treatment modalities, and prognosis differ between the two pathologies. The classification criteria based on histology, such as MM and TNM criteria, do not fully meet the clinical requirements. The patient would mainly undergo curative resection if the multiple lesions are classified as MPLC. In contrast, the patient would more likely benefit from systemic treatment modalities, such as chemotherapy or targeted therapy, if IPM is present (Asamura, 2010). Samples that cannot be differentiated based on standard criteria can be identified using molecular methods. Donfrancesco et al. (2020) reported that pathologic criteria seem to be less accurate than molecular criteria for staging multiple lung adenocarcinomas, suggesting that the pathologic features can be combined with molecular analysis for accurate diagnosis. Bruehl et al. (2022) indicated that a subset of lung adenocarcinomas with two independent nodules (judged relevant using histologic assessment) can be proved unrelated using molecular testing for common driver mutations, leading to downstaging. A meta-analysis suggested that the molecular methods represent the gold standard for identification when the predominant architecture pattern is evident in tumor but different patterns are observed in heterogeneous cases or small samples (Mlika, Zorgati & Mezni, 2020). However, the available molecular identification criteria are not detailed enough to accurately distinguish MPLC and IPM. In our study, 35 patients were divided into training and validation cohorts, and comprehensive molecular identification criteria were proposed by comparing them with the previously proposed molecular criteria to distinguish MPLC and IPM.

The concordance rate of gene mutations in the primary tumor and metastasis is >90%, whereas that in multiple lung tumors (e.g., MPLC) is 10.3–32.6%. We found that patients with MPLC did not have any trunk or shared mutation or had fewer than two. However, 98.02% of the mutations observed were branch mutations, indicating that 69.23% of MPLC did not have any mutations in common. In contrast, a higher number of trunk or shared mutations were found in IPM, indicating significant differences among the mutant components. Several authors have reported approximately 50% prevalence of EGFR mutations in the East Asian populations. Among these mutations, EGFR p. L858R and EGFR p. 19del are the two most commonly observed mutations, accounting for approximately 20% of mutations found in patients with lung adenocarcinoma (Shi et al., 2014; Zhou et al., 2016). Consistent with previous studies (Zhao et al., 2018), EGFR was the most commonly mutated gene in lung cancer. In our study, the EGFR gene alterations were identified in 38 lesions from 21 patients (60%) enrolled in the study. Among the MPLC lesions, 16 lesions showed EGFR mutations and eight lesions showed EGFR p. L858R (22.7%). Among the IPM lesions, 23 lesions from 10 patients showed EGFR alterations, and 12 lesions from six patients (46.2%) showed EGFR p. L858R. The prevalence of EGFR p. L858R in patients with MPLC was found to be lower compared with that in patients with IPM. This observation indicates a negative selection for EGFR p. L858R mutations in MPLC. Furthermore, we analyzed the number of EGFR, KRAS, TP53, BRAF, ERBB2, MET, RET, NRAS, ALK, and ROS1 driver gene mutations in patients identified with MPLC and IPM. Mutations of certain driver genes, such as EGFR, KRAS, and TP53, have been recommended as potential clonal markers in MPLC (Qiu et al., 2020; Wu et al., 2013). The RAS proteins are involved in the mitogen-activated protein kinase and phosphatidylinositol 3-kinase signaling pathways and are implicated in regulating cell growth, survival, and differentiation (Chen et al., 2021; Malumbres & Barbacid, 2003). EGFR mutations are found in approximately 50% of cases recorded in Asian populations, whereas KRAS mutations predominate in Caucasian lung cancer populations with a frequency of 30–35% (Reck et al., 2021). KRAS is prevalent in various regions of lung cancer tissues in Caucasian populations, even after subregional molecular testing of patient tumor tissues after sectioning. MPLC is dominated by KRAS mutations in European and American populations (Chang et al., 2019; Mansuet-Lupo et al., 2019), likely reflecting the overall known geographic differences in genomic profiles of lung cancer. Our results demonstrated that the prevalence of EGFR (52.38% vs. 71.43%; p = 0.31) and TP53 (42.86% vs. 57.14%, p = 0.5) mutations was lower in MPLC compared with that in IPM. In contrast, the prevalence of KRAS mutation (42.86% vs. 7.14%; p = 0.03) was higher in MPLC compared with that in IPM, indicating significant differences between the mutant components in MPLC and IPM.

Tumor classification based on imaging and pathologic features can provide the basis for staging and guiding subsequent treatment (Detterbeck et al., 2016). However, no clear criteria are available to distinguish MPLC and IPM if independent nodules of the same pathologic type exist (Martini & Melamed, 1975). Therefore, in-depth evaluation is critical to determine whether patients with multiple lesions can be categorized as MPLC or IPM. MPLC is characterized by considerable heterogeneity because multiple primary lesions often do not share any or only have a few common changes. While individual lesions lacking common mutations are easily identified as MPLC, thorough scrutiny and interpretation are necessary for multiple lesions with shared alterations. Therefore, additional molecular evidence is essential to distinguishing MPLC and IPM. Shao et al. (2020) proposed that the tumors that share at least two gene mutations or have one rare mutation in common can be classified as IPM. Mansuet-Lupo et al. (2019) proposed that IPM can be interpreted when all alterations are common between tumors or at least one mutation is shared between tumors with one or more additional alterations in one of the samples. Wang et al. (2020) suggested that IPM can be interpreted if more than three common alterations are found. Pei et al. (2021) proposed that IPM can be identified when the same driver gene mutation (exclusive to EGFR L858R) is shared between tumors or when all alterations are common between the tumors in the patient. Our training cohort patients were identified according to the currently proposed molecular criteria for distinguishing MPLC and IPM. We found that the diagnosis of 14/ 22 (63.6%) patients was consistent using the later three molecular criteria. Patients 5, 6, 8, 12, and 15 were identified using the previously proposed molecular methods but had different classifications. Patient 5 (common mutations: ERBB2 c.1022-6C > T and ROS1 fusion) and patient 8 (common mutations: EGFR p. L858R and TP53 p. W146*) were identified as IPM based on the molecular methods proposed by Mansuet-Lupo et al. (2019) (IPM can be interpreted when tumors have at least two mutations in common) and Pei et al. (2021) (IPM can be identified when EGFR p. L858R and another mutation are shared between tumors or consistent gene hotspot mutations (EGFR, KRAS, BRAF, ERBB2, ALK, ROS1, MET, or RET) in tumors but do not share EGFR p. L858R). However, these two patients were identified as MPLC based on the molecular criteria proposed by Wang et al. (2020) i.e., lesions can be classified as MPLC if one TKI-related driver gene alteration (EGFR, ALK, ROS1, MET, RET, BRAF, and KRAS) and one non-TKI-related gene alteration are present (with a total of common mutations ≤2). Moreover, MPLC was confirmed as the final classification based on the pathologic features of patients and molecular identification. The PIK3CG p. H199N and POLD1 p. A532T mutations were common in the two tumors of patient 12. The patient was identified as IPM based on the criteria proposed by Mansuet-Lupo et al. (2019) and Pei et al. (2021). However, the interpretation was MPLC based on the criteria proposed by Wang et al. (2020). We finally identified the two lesions as MPLC based on the histologic types (AIS and MIA) and the common mutations. Notably, all the molecular identification criteria were different when only two common mutations were identified in all tumors. In our study, we focused on adenocarcinoma and introduced a more accurate and comprehensive method compared with previous methods using NGS to differentiate MPLC from IPM. We recruited 13 patients (with 29 samples) and identified 8 MPLC and 5 IPM cases, and the identification was in agreement with the results of independent imaging and pathologic classification. Moreover, we reported the cases of four Chinese patients for whom targeted NGS profiling provided key additional information that allowed the definitive identification of MPLC or IPM.

Despite the low mutation rate, backbone mutations occurred in two cases of MPLC. This may occur because the molecular characteristics of MPLC and IPM are not completely exclusive. Therefore, molecular features combined with histology are not adequate for clinical classification. The importance of obtaining a pre-resection tissue diagnosis of all lesions is emphasized in the separation of multiple synchronous primary tumors from intrapulmonary metastases in the eighth edition of the 8th TNM classification for lung cancer (Waller, 2018). This classification indicates surgical resection in all categories of multiple lesions; however, the surgical strategy is different in each case. Nevertheless, there are some important overriding principles (Waller, 2018). We suggest that these types of patients require additional follow-up in clinical management to avoid the lesion being a potential IPM and missing the optimal time for intervention. The molecular variant concordance of MPLC and IPM is not completely mutually exclusive. Histologic classification methods are not feasible when rare variants are present or all samples are mutation-free.

Compared with previous studies, our model assessed the impact of the number of tumor samples and the rate of molecular variant concordance between samples on classification. We extended the number of genes examined to 450 compared with 22 genes considered in an earlier study (Mansuet-Lupo et al., 2019). This allowed us to examine the effect of certain uncommon mutations on classification and precisely examine the rate of molecular variant agreement between samples. We performed a more detailed analysis of samples with molecular variant concordance of 0%, considering the effect of the number of tumor samples compared with the previously reported analyses (Pei et al., 2021; Wang et al., 2020). Classification could not be determined if the molecular variance between samples = 0% and no variance was observed in all samples. Samples were classified as MPLC if the total number of samples was ≤2 and no mutation was observed in one of the tumor samples and a single mutation in the other sample. However, the classification could not be determined if the total number of samples was >2 and no mutation was observed in one of the samples. In addition, we could not determine the classification when >0% of the intersample agreement in molecular variations was due to two uncommon mutations. This is because the uncommon mutations are not oncogenic driver genes, and accurate judgments of oncogenicity and prognosis cannot be made. Notably, our model and others have considered the important role of driver genes for classification, suggesting that multiple samples with the same driver gene can be classified as IPM.

To conclude, identifying genetic alterations may be an effective method for classifying MPLC and IPM. Here, we proposed comprehensive molecular criteria based on the previously published molecular identification criteria of MPLC and IPM, and our criteria may be useful for differential classification. Although our new molecular criteria accurately classified MPLC and IPM, several limitations of the present study should be acknowledged. First, our study sample was limited to Chinese patients with typical mutational features of the Asian lung cancer populations. The applicability of this classification model to the Caucasian population, where KRAS mutations predominate, needs to be further examined. Second, our typing strategy is ineffective for samples without mutations. These samples may need to draw on other types of data in addition to mutations, such as transcriptomic or even immune-related features, which would require a higher quantity and quality of samples (Song et al., 2023). In addition, sampling errors due to tumor heterogeneity can be avoided by multiregional testing. In addition, the accuracy of the classification can be increased using whole-genome sequencing, which comprehensively examines the consistency of genetic variants in terms of copy number instability and genome-wide amplification (Frankell et al., 2023). Overall, more recommendations from official organizations are needed to guide the use of molecular markers in defining nodal clonality in multiple lung tumors.

Supplemental Information

Supplemental Information 1 Mutation status and concordance rate for lesions of the training cohort.

(A) The ratio of trunk, shared, and branch mutations are shown in MPLC and IPM of training cohort. (B) The ratio of trunk, shared, and branch mutations are shown in MPLC and IPM of validating cohort.

Click here for additional data file.

Supplemental Information 2 Supplementary Tables.

Click here for additional data file.

Supplemental Information 3 TZQ-MPLC and IPM data.

Click here for additional data file.

The authors would like to thank Shaohua Yuan for performing genomic profiling.

Additional Information and Declarations

Competing Interests

Author Contributions

Human Ethics

Data Availability

Lijie Guo and Fanfei Meng are employees of OrigiMed.

Zhenhua Li conceived and designed the experiments, performed the experiments, prepared figures and/or tables, authored or reviewed drafts of the article, and approved the final draft.

Huilai Lv conceived and designed the experiments, analyzed the data, authored or reviewed drafts of the article, and approved the final draft.

Fan Zhang performed the experiments, authored or reviewed drafts of the article, and approved the final draft.

Ziming Zhu performed the experiments, prepared figures and/or tables, and approved the final draft.

Qiang Guo performed the experiments, prepared figures and/or tables, and approved the final draft.

Mingbo Wang performed the experiments, prepared figures and/or tables, and approved the final draft.

Chao Huang analyzed the data, prepared figures and/or tables, and approved the final draft.

Lijie Guo analyzed the data, authored or reviewed drafts of the article, and approved the final draft.

Fanfei Meng analyzed the data, authored or reviewed drafts of the article, and approved the final draft.

Ziqiang Tian conceived and designed the experiments, analyzed the data, prepared figures and/or tables, authored or reviewed drafts of the article, and approved the final draft.

The following information was supplied relating to ethical approvals (i.e., approving body and any reference numbers):

The studies involving human participants were reviewed and approved by Ethics approval was granted by the Ethics Committee of the Fourth Hospital of Hebei Medical University. Written informed consent to participate in this study was provided by the participants’ legal guardian/next of kin.

The following information was supplied regarding data availability:

The raw data are available at figshare: Tian, Ziqiang (2023). Proposal of criteria to distinguish multiple primary lung cancer from intrapulmonary metastasis in patients with multiple lung tumors. figshare. Dataset. https://doi.org/10.6084/m9.figshare.24164883.v1.

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
