# Peer review of "Using molecular characteristics to distinguish multiple primary lung cancers and intrapulmonary metastases"

_PeerJ, doi:10.7717/peerj.16808_

## Round 0.1 · original submission · Major Revisions

Both reviewers raised many questions and suggestions. Especially regarding experimental design, they all gave specific points that needed attention. I suggest you take these comments seriously and respond and revise them one by one.

**Language Note:** PeerJ staff have identified that the English language needs to be improved. When you prepare your next revision, please either (i) have a colleague who is proficient in English and familiar with the subject matter review your manuscript, or (ii) contact a professional editing service to review your manuscript. PeerJ can provide language editing services - you can contact us at [email protected] for pricing (be sure to provide your manuscript number and title). – PeerJ Staff

Reviewer 1 ·

Basic reporting

1) The distinction between intrapulmonary metastases (IPM) and multiple primary lung cancer (MPLC) in patients with multiple lung cancers is currently challenging and crucial. The theme of this study is to propose a more comprehensive and detailed approach to differentiate between IPM and MPLC. The study focused on clinical samples, and a more in-depth bioinformatics analysis of the data obtained by next-generation sequencing (NGS) analysis established the genomic mutational differences between IPM and MPLC, and correlated the differences with imaging and pathologic diagnosis. The differential characterization between IPM and MPLC revealed in this study will provide a basis for clinical lung cancer diagnosis. Overall, this study is logical and meets the requirements for publication. However, It is suggested that a round of careful proofreading in grammar and diction will be more helpful for readers to understand the research content.
2) Although the reference is not obviously inappropriate, more attention should be paid to the recent research progress in the field. The author even did not cite the references in the past two years at all, which needs to be further improved.

Experimental design

1) The author state in line 65-68 that there are three studies that reveal ways to differentiate between IPM and MPLC, please expand on the details of these three studies as well as their flaws to clarify the significance of this study.
2) In the Introduction section, it is suggested that after stating the definitions of MPLC and IPM, add the reason why the two are to be differentiated, stating is it because of a difference in treatment? Or is there a difference in the development of the two?
3) line 72-73 suggests that the molecular criteria constructed in this study are more detailed and comprehensive, please describe in what ways this is reflected and explain the advantages of the criteria revealed in this study that distinguish them from other known criteria.
4) The "Study Design" describes "Imaging findings, anatomical evidence, pathological findings, histological examination of the primary tumor, and histological subtypes" to differentiate between MPLC and IPM, but does not specify what data these include and how the samples were analyzed by these bases. In the "Study Design", it describes "Imaging findings, anatomical evidence, pathological findings, histological examination of the primary tumor, and histological subtypes" to differentiate between MPLC and IPM, but it does not specify what data are included in these bases and how the samples were analyzed by these bases, so please add a detailed description.
5) In line 159-187 it is clarified that this study utilized previously reported methods to classify the samples in this paper, and that the conclusions of these methods of classification lack a summarizing statement, please add the relevant statement, in particular to describe what are the commonalities and differences in the classification of the samples by these methods.

Validity of the findings

1) The results of Figure 2 need to be analyzed in depth, with a focus on elucidating what differences and commonalities there are between the mutational features of MPLC and those of IPS, and in particular, how these commonalities can be circumvented to maximize the differences between the two when feature differentiation is performed at a later stage.
2) In Figure 3C, despite the low mutation rate, trunk mutation occurred in 2 cases of MPLC, how should this situation be circumvented in the follow-up? Please point it out in the discussion section.
3) In the Discussion section of the description, how do high-resolution imaging devices relate to molecular detection modalities? Why is it claimed that the increased detection of multiple pulmonary nodules prompted by imaging devices necessitates the development of new diagnostic methods? Please check the original article and revise the presentation or add clarification as necessary.
4) It is suggested that more detailed relevant studies on KRAS mutations be added to the Discussion section, such as link KRAS mutations to the development of lung cancer, which in turn leads to the predominance of KRAS mutations in MPLC in the European and American populations.
5) How can it be ascertained that the method provided in this paper is accurate for the distinction between MOLC and IPM? How can it be determined that the method is superior to the other methods mentioned in this paper? Please summarize this in the discussion section.

Reviewer 2 ·

Basic reporting

The proposed comprehensive molecular criteria for differentiating between MPLC and IPM in patients with multiple lung tumors showed high accuracy and alignment with independent imaging and pathological diagnoses. The analysis of genomic profiles using targeted NGS revealed distinct mutation patterns, with MPLC showing mostly branch mutations and IPM showing a higher percentage of trunk mutations and shared mutations. These findings provide valuable insights into the understanding and diagnosis of MPLC and IPM, allowing for more accurate and tailored treatment strategies for patients with multiple lung tumors. The writing structure of the manuscript is clear, which can make people understand the intention, and the original data is relatively reasonable and complete. However, the article's references lack a survey of the latest progress, especially the literature citations in the last three years need to be increased.

Experimental design

1. The manuscript titled “Proposal of criteria to distinguish multiple primary lung cancer from intrapulmonary metastasis in patients with multiple lung tumors”. It is better to reflect the findings of this study in the title.
2. Lung cancer is a heterogeneous disease, and there can be significant variability in molecular profiles among different patients. The proposed criteria focused on a 450-gene targeted NGS panel, but it is possible that other molecular alterations or markers may be relevant for distinguishing between MPLC and IPM. The study would benefit from a more comprehensive analysis of additional molecular markers.
3. The Introduction section mainly introduces the continuous exploration of identification criteria for IPM and MPLC, but does not mention the background reasons for distinguishing IPM and MPLC, such as their clinical implications in treatment and prognosis
4. The criteria proposed in the study rely on the interpretation and analysis of genomic profiles. However, there can be inherent inter-observer variability in the interpretation of genetic alterations, which may affect the consistency and reproducibility of the results. It would be valuable to assess the inter-observer agreement in the application of the proposed criteria.
5. The objective of the study was that “Differentiation between intrapulmonary metastasis (IPM) and multiple primary lung cancers (MPLC) in patients with multiple lung tumors is critical but challenging. We aimed to propose a more comprehensive and detailed molecular criteria of MPLC and IPM”. It is necessary to directly clarify the current challenges and then explain why molecular standards should be established.
6. The patients included in the study were selected based on their eligibility to undergo genomic profiling using targeted NGS. This may introduce selection bias, as patients with certain characteristics or tumor types may have been more likely to be included in the study, potentially influencing the results and the ability to generalize the findings to a broader population. In addition, the author should explain how to group training cohort and validation cohort as Twenty-two lung cancer patients (each patient ≥ 2 tumors) were enrolled in a training cohort, and 13 patients were recruited as the validation cohort.
7. The author described that “Histological classification and clinical data alone are not sufficiently satisfactory to distinguish MPLC from IPM, so molecular profiling is used to assist in further confirmation of the distinction in such cases. At present, we found that there are three research teams proposed molecular identification criteria for distinguishing MPLC from IPM, including Audrey Mansuet-Lupo et al., Xiaohui Wang et al. and Guotian Pei et al. which provides a more effective identification tool for distinguishing between MPLC and IPM. However, the proposed molecular identification criteria are not detailed enough and also have some limitations in distinguishing the accuracy of MPLC and IPM.” In the Introduction. The author needs to elaborate on why it is not detailed enough and what are the limitations of accuracy n distinguishing MPLC and IPM. References need to be cited in the article.
8. Materials and methods need to be described in detail, such as “Sample preparation, targeted NGS, and genetic analysis”.

Validity of the findings

1.Some descriptions in the article are inaccurate and require full text verification. Such as “In conclusion, Genetic alterations may be an effective method for the diagnosis of MPLC and IPM, and NGS may serve as a useful tool to assist differential diagnosis.”. The focus of this study is not the application of NGS in assisting differential diagnosis, especially in diagnosis of MPLC and IPM.
2.It is necessary to check the description of each part of the result to ensure that it is corresponding and accurate.
3. Add more comparisons with the currently established identification methods in the Discussion section.
4.In the study, molecular methods and Martini and Melamed criteria were used to identify whether patients belonged to MPLC or IPM, and the consistency of the identification results between the two modalities was analyzed. The provided information illustrates the utilization of both molecular criteria and Martini and Melamed criteria in the characterization of MPLC and IPM for these specific patients. These results highlight the potential for these approaches to provide valuable insights into the distinction between MPLC and IPM by integrating molecular profiling with clinical and pathological information.
5.Overall, while the study provides important insights into the molecular characterization of MPLC and IPM, there are several limitations that need to be considered as discussed in the manuscript. The author needs to discuss the application of this method in other diseases and the possibility of future application in distinguishing MPLC and IPM.

---

## Round 0.2 · accepted · Accept

Both reviewers have found your study to be of high quality and have recommended its acceptance for publication. We appreciate your contribution to our journal and look forward to publishing your work.

The reference Martini & Melamed 1975 should be moved to the first sentence cited.

Reviewer 1 ·

Basic reporting

no comment

Experimental design

no comment

Validity of the findings

no comment

Additional comments

Revised manuscripts may be accepted in peerJ.

Reviewer 2 ·

Basic reporting

The author's revision is effective and basically solves my concerns.

Experimental design

no comment

Validity of the findings

no comment